# Synthesis of ZnO/Au Nanocomposite for Antibacterial Applications

**DOI:** 10.3390/nano12213832

**Published:** 2022-10-30

**Authors:** Violeta Dediu, Mariana Busila, Vasilica Tucureanu, Florentina Ionela Bucur, Florina Silvia Iliescu, Oana Brincoveanu, Ciprian Iliescu

**Affiliations:** 1National Research and Development Institute in Microtechnologies—IMT Bucharest, 126A Erou Iancu Nicolae Street, 077190 Bucharest, Romania; 2Centre of Nanostructures and Functional Materials-CNMF, “Dunarea de Jos” University of Galati, Domneasca Street 111, 800201 Galati, Romania; 3Faculty of Food Science and Engineering, “Dunarea de Jos University” of Galati, Domneasca Street 111, 800201 Galati, Romania; 4Faculty of Chemical Engineering and Biotechnologies, University “Politehnica” of Bucharest, 011061 Bucharest, Romania; 5Academy of Romanian Scientists, 010071 Bucharest, Romania; 6Regional Institute of Oncology, Iasi TRANSCEND Research Center, 2-4 General Henri Mathias Berthelot, 700483 Iasi, Romania

**Keywords:** ZnO/Au, small nanorods, solvothermal method, antimicrobial materials, *E. coli*, *S. aureus*

## Abstract

Annually, antimicrobial-resistant infections-related mortality worldwide accelerates due to the increased use of antibiotics during the coronavirus pandemic and the antimicrobial resistance, which grows exponentially, and disproportionately to the current rate of development of new antibiotics. Nanoparticles can be an alternative to the current therapeutic approach against multi-drug resistance microorganisms caused infections. The motivation behind this work was to find a superior antibacterial nanomaterial, which can be efficient, biocompatible, and stable in time. This study evaluated the antibacterial activity of ZnO-based nanomaterials with different morphologies, synthesized through the solvothermal method and further modified with Au nanoparticles through wet chemical reduction. The structure, crystallinity, and morphology of ZnO and ZnO/Au nanomaterials have been investigated with XRD, SEM, TEM, DLS, and FTIR spectroscopy. The antibacterial effect of unmodified ZnO and ZnO/Au nanomaterials against *Escherichia coli* and *Staphylococcus aureus* was investigated through disc diffusion and tetrazolium/formazan (TTC) assays. The results showed that the proposed nanomaterials exhibited significant antibacterial effects on the Gram-positive and Gram-negative bacteria. Furthermore, ZnO nanorods with diameters smaller than 50 nm showed better antibacterial activity than ZnO nanorods with larger dimensions. The antibacterial efficiency against *Escherichia coli* and *Staphylococcus aureus* improved considerably by adding 0.2% (*w*/*w*) Au to ZnO nanorods. The results indicated the new materials’ potential for antibacterial applications.

## 1. Introduction

Specific Gram-positive and -negative pathogenic bacteria responsible for threatening human infections were treated successfully with antimicrobial chemotherapy, significantly improving the average life expectancy and the quality of life over the past decades. However, improper antibiotics utilization led to new generations of super-bacteria resistant to a large spectrum of antibiotics [1]. In most cases, treating the infections caused by antibiotic-resistant bacteria is not only costly but requires toxic alternatives, sometimes ineffective, risky, and associated with higher mortality.

WHO declared antimicrobial resistance (AMR) one of the ten global health issues [2]. Reports stated that the antimicrobial-resistant infections related mortality caused in 2016 about 700,000 deaths worldwide [3], with a predicted accelerated growth trend due to the increased use of antibiotics during the coronavirus pandemic and the antimicrobial resistance, which grows exponentially, disproportionately to the current rate of development of new antibiotics. A 10 million per year death rate linked to antimicrobial resistance is expected by 2050 [4]. Consequently, addressing the mounting risks should be a 21st-century public health priority aiming at searching for new materials with antimicrobial properties to counterbalance the decline in the development of new antibiotics. One of the approaches is using nanomaterials with different structures and chemical compositions [5]. For instance, the metal oxide nanoparticles (NPs) intensively investigated over the last two decades, proved to be a new line of defense against multi-drug resistance microorganisms. The advantage of using NPs-based materials with antibacterial activity over conventional antibiotics is a better ratio between therapeutic efficacy and the side effects. Over the years, several mechanisms of action (MOA) have been proposed for the NPs’ antibacterial action: metal ion release [6], induction of oxidative stress by the free radical formation that can either damage cellular components or inhibits enzyme activity and DNA synthesis [7,8]. Due to the variety of NPs’ MOA, sometimes one type of NP presents different simultaneous MOA, making it difficult for bacteria to become resistant to NPs, which would involve multiple simultaneous gene mutations [9]. NPs’ MOA involves either the intrinsic antibacterial properties, acting directly on the pathogenic bacteria or the ability to act as nanocarriers (e.g., liposomes, polymeric NPs, and solid lipid NPs) for the targeted delivery of antibiotics in tissue or cell [10]. Within a biological medium, the NPs interact physio-chemically with the bacteria and express either bactericidal or bacteriostatic actions. Although their activity is slower than the organic antibacterial agents, NPs do not release dangerous by-products [11]. The bactericidal activity of NPs depends on composition, size, shape, crystallinity, defects, surface modification, and concentration in the culture media and the type of bacteria (Gram-positive or Gram-negative species) [12]. Consequently, different inorganic NPs have been extensively studied for antibacterial treatment such as Ag [13,14], ZnO [6,15,16], Cu/CuO [17], TiO_2_ [18], Fe_3_O_4_ [19], SiO_2_ [20], carbon nanotubes [21], Au-Ag core-shell NPs [22]. In addition, the bactericidal NPs may be combined with polymers and coated onto different functional surfaces such as medical instruments and devices to further extend the range of commercial applications [23].

Among these nanomaterials, ZnO has been explored extensively and proved antibacterial activity against a wide range of (mostly Gram-positive) bacteria strains, generally more efficient than other NPs even at low concentrations. In addition, ZnO is a biocompatible, inexpensive, and earth-abundant material. Furthermore, compared to widely used Ag NPs, this oxide has some practical advantages, such as low production cost and photochemical stability. Different ZnO based nanostructures have been tested for antibacterial response such as NPs [24,25,26], nanorods [27] nanoflowers [28], nanofibers [29], Ag doped ZnO NPs [30,31], surface decorated NPs [32], nanocomposites [33,34], and surface modified NPs [35]. Ag NPs are considered a classic antimicrobial material that has proven to be very effective against a large variety of microbes [36,37]. Ag-based nanocomposites with enhanced antimicrobial properties have been developed [38,39].

Au NPs are less studied for their antibacterial effect than silver and ZnO, and there is little information on how these NPs act. Their unique physicochemical properties may facilitate the Au NPs’ capacity as antibacterial agents [40,41,42]. Au nanomaterials can also act through a photothermal-triggered mechanism, Au converting photon energy into heat under near-infrared light, a phenomenon called localized surface plasmon resonance [43,44,45]. Furthermore, the antibacterial efficiency against drug-resistant strains can be extended to an extensive pH range, as in the case of the zwitterion-modified Au nanorods, due to their pH-responsive surface charge transition activities [43]. Coating Au nanorods or nanostars with polyelectrolytes improves the antibacterial rates through a chemo-photothermal synergistic effect and confers biocompatibility and stability [44]. 

Although ZnO and Au NPs were tested independently against different bacteria, a few papers reported combinations of the two NPs against bacterial pathogens: Au decorated ZnO NPs [31,46].

Here we report the synthesis and characterization of different ZnO and ZnO/Au nanomaterials to evaluate their antibacterial activities, the impact of their structure and morphology on antimicrobial performance, and to propose the most suitable antimicrobial mechanism against *Staphylococcus aureus* (Gram-positive bacteria causing skin and soft tissue and bloodstream infections), and *Escherichia coli* (Gram-negative bacteria responsible for common gastrointestinal tracts infections and extraintestinal infections which can even lead to death). The obtained ZnO and ZnO/Au nanomaterials may find potential applications as antimicrobial coatings in the healthcare industry.

## 2. Materials and Methods

### 2.1. Materials

Zinc nitrate hexahydrate (Zn(NO_3_)_2_∙6H_2_O) 98%, sodium dodecylbenzenesulfonic acid salt—SDBS (CH_3_(CH_2_)_11_C_6_H_4_SO_3_Na) technical grade, potassium hydroxide (KOH) ≥85%, ethanol—C_2_H_5_OH (98%) L-lysine H_2_N(CH_2_)_4_CH(NH_2_)CO_2_H ≥98%, Au (III) chloride trihydrate (HAuCl_4_∙3H_2_O) ≥99.9%, sodium borohydride (NaBH_4_) 98%, and Luria-Bertani broth (LB) were purchased from Sigma-Aldrich (Darmstadt, Germany) and Tryptone soy broth (TSB) from Oxoid L.T.D.( Basingstoke, Hampshire, England). Gram-positive bacteria—*Staphylococcus aureus* ATCC 25923 and Gram-negative bacteria—*Escherichia coli* ATCC 25922 were provided by the Laboratory of Foods Physico-Chemical and Microbiological Analysis, Dunarea de Jos University (Galati, Romania). 

### 2.2. ZnO and Zn/Au Synthesis

The antibacterial nanomaterials were obtained by a two-stage procedure, illustrated in Figure 1. ZnO nanostructures were obtained by solvothermal method without a seeding step, starting from a modified method reported in [47]. The synthesis of ZnO/Au involved the formation of Au NPs on the surface of already obtained ZnO nanorods, using L-lysine to restrict the growth of Au NPs and ensure their uniform deposition on the surface of ZnO nanorods. Au NPs were obtained through the Brust-Schiffrin method, starting from chloroauric acid, using sodium borohydride as a reducing agent. 

#### 2.2.1. ZnO Synthesis

(1.a) In a Berzelius beaker containing 30 mL of ethanol and 10 mL of deionized water, 0.8925 g of zinc nitrate hexahydrate was dissolved by stirring for 20 min. Next, 1.0425 g of SDBS and 3 g of potassium hydroxide were added. The solution was stirred for 30 min, transferred to a 45 mL Teflon-lined stainless-steel pressure vessel (autoclave) and placed in the preheated oven at 100 °C for 10 h. After cooling to room temperature, the solution was centrifuged. Then the obtained powder was washed several times with ethanol and deionized water. The resulting product was oven-dried at 60 °C for 12 h. This sample was coded Z1.

(1.b) Another synthesis was performed following the same procedure as in (1.a) but without the addition of deionized water. In addition, this sample was washed only with ethanol. The sample was coded Z2.

#### 2.2.2. ZnO/Au Synthesis

(2.a) 60 mL of 0.05 mM aqueous solution of chloroauric acid and 0.3 g of ZnO (obtained in step (1.a) were mixed in a glass vessel maintained at 6 °C. 2 mL of 0.01 M L-lysine aqueous solution was added to this mixture under stirring. 0.1 mM sodium borohydride aqueous solution kept at 6 °C in 0.1 mL portions was added until the color changed from white to faint purple, and the magnetic stirring continued for 30 min. The resulting suspension was left for 16 h and then centrifuged. The residue was washed repeatedly with ethanol and water and dried in an oven at 60 °C for 8 h. This sample was codded ZA1.

(2.b) In this synthesis path, the addition of Au NPs to ZnO (obtained in (1.b) takes place in the same way as in (2.a) but avoiding water. The ethanol replaced water in chloroauric acid and sodium borohydride solutions, and ethylene glycol was used to dissolve L-lysine. In this case, the resulting precipitate has a light ruby red color. This sample was codded ZA2.

### 2.3. Characterisation Techniques

The crystallinity of the nanomaterials was further assessed by X-Ray diffraction (XRD) on a SmartLab X-Ray Diffractometer (Rigaku Americas Corporation, Tokyo, Japan) using CuKα radiation (λ = 1.5406 Å). The phase identification was made by referring to the International Center for Diffraction Data (ICDD). The resulting nanostructures morphology was examined using an Quanta Inspect F scanning electron microscope (FEI Company, Eindhoven, The Netherlands), (SEM) coupled with energy dispersive X-rays spectroscopy (EDX). The structure, size and crystalline phase were investigated with transmission electron microscopy (TEM) with a JEOL JEM-2011 FasTEM (JEOL Ltd., Tokyo, Japan). A suspension of the NPs (0.1 mg) in isopropanol (10 mL) was sonicated for 15 min, and a drop was placed onto a copper grid. The FTIR spectra were recorded on a Tensor 27 Bruker Optics spectrometer using an ATR Platinum holder (Bruker Corporation, Billerica, MA, USA). The specimens were pressed into the pellets using spectroscopically pure KBr as a matrix. Hydrodynamic diameters and surface electrical charge were determined through dynamic light scattering (DLS) techniques using a Beckman-Coulter Delsa™Nano Submicron Particle Size and Zeta Potential Analyzer (Beckman Coulter Inc., Brea, CA, USA). The concentrations of Zn and Au in synthesized nanomaterials were determined by inductively coupled plasma optical emission spectrometry (ICP-OES) Optima 5300 DV (Perkin Elmer Inc., Waltham, MA, USA). The absorbance in visible light of resulted formazan (for the antibacterial test) was measured using a Cary 5E-Varian UV-VIS-NIR spectrophotometer (Agilent Technologies Inc., Santa Clara, CA, USA).

### 2.4. Antibacterial Activity

#### 2.4.1. Disc Diffusion Assay

*S. aureus* and *E. coli* ATCC 25922 were stored in tryptone soy broth (TSB) and Luria-Bertani broth (LB), respectively, containing 30 % (*v*/*v*) glycerol at −20 °C. Before use, stock cultures of both types of bacteria were activated by transferring twice in TSB and LB, respectively. The bacteria strains were grown at 37 °C for 18 h to obtain the working cultures. The newly prepared cultures were diluted to an optical density (OD 600) of 0.1 in a fresh culture medium and grown to 0.5 and 0.4 OD 600, respectively, to yield a bacterial concentration of approximately 10^8^ CFU/mL. Bacterial cultures were then diluted to 10^6^ CFU/mL, and 100 µL of each were evenly spread on Mueller-Hinton Agar plates (MHA; Oxoid, Basingstoke, Hampshire, England) with a sterile dispensable Drigalski spatula. Antimicrobial susceptibility testing discs (Bioanalyse, Ankara, Turkey) were impregnated with 10 µL of ZnO or ZnO/Au suspensions, allowed to dry for 10 min and placed in triplicate on the seeded MHA plates. The testing plates were incubated at 37 °C for 12 h, and the inhibition zones’ diameters were measured. For a fair comparison, our study used a concentration value within the range of concentrations applied to test of antibiotics in antibiograms.

#### 2.4.2. Tetrazolium/Formazan (TTC) Assay

1 mL of each ZnO and ZnO/Au dispersion (5 mg/mL) were placed in flasks with 40 mL nutrient broth containing 10 μL of cell/mL (10^8^ CFU/mL). The flasks were incubated while agitating (200 rpm) for 3 h at 37 °C. 1 mL from the control sample, and experimental samples were added into sterilized testing tubes containing 100 μL TTC (0.5 %, *w*/*v*), then incubated at 37 °C for 20 min. The resulting formazan from the reduction of 2,3,5,-triphenyl tetrazolium chloride in the presence of *E. coli* or *S. aureus* bacteria [48] was separated by centrifugation and then washed with ethanol. The positive control consisted of a bacterial suspension in an appropriate culture medium with an amount of ethanol corresponding to the highest quantity in the broth microdilution assay. The absorbance value of formazan at 480 nm determined cellular activity and viability. 

### 2.5. Statistical Analysis 

The experimental results were analyzed using descriptive statistical techniques such as mean and standard deviation. ImageJ software (University of Wisconsin, Madison, WI, USA) was used for processing and analyzing SEM images, and the histogram distributions were then fitted to the Gaussian function using OriginPro 8.5 data processing software( OriginLab, Northampton, MA, USA) by Gaussian function. One-way ANOVA was employed to process the antimicrobial results, and data analysis was performed using Microsoft Excel (Microsoft Corp., Redmond, WA, USA). *p* value < 0.05 was considered as significant. 

## 3. Results and Discussion

### 3.1. Structure and Morphology of ZnO Nanostructures

XRD performed structural characterization on obtained nanomaterials (Figure 2). The diffraction peak positions were well attributed to a single-phase wurtzite structure (hexagonal phase, space group *P63mc*) for all the samples. The XRD patterns are in good agreement with JCPDS-card No. 36-1451 [49].

In all the samples, the (1 0 1) peak of the ZnO crystal is the most intense, meaning it is the preferred growth plane. The calculated lattice constants of this hexagonal phase are a = b = 3.2529(8) Å and c = 5.2075(2) Å, for sample Z1, and a = b = 3.2525(6) Å and c = 5.2018(9) Å, for ZA1. Comparing the XRD pattern of Au/ZnO with that of the ZnO sample in Figure 2 showed they are similar, meaning that the formation of Au during the reduction reaction did not noticeable influence the crystal structure of ZnO. The strain in the wurtzite lattice is 0.33 for sample Z1 and slightly increased to 0.35 in the case of the ZA1 sample. In the XRD pattern of the ZA1 sample, small additional diffraction peaks appeared at ca. 38.36°, 44.37°, 64.65°, and 81.41°, which are ascribed to the (111), (200), (220), and (222) planes of face-centered cubic (fcc) crystal cell of Au (JCPDS No. 01-1172) (which has a concentration of about 0.2 % (*w*/*w*) in ZA1 or ZA2 materials, value resulted from ICP OES analysis). No impurities were observed in the XRD patterns of the Z1 and ZA1 samples.

In the case of samples obtained using only ethanol as a solvent, the wurtzite ZnO lattice constants are a = b = 3.2539(6) Å and c = 5.2161(10) Å for the Z2 sample and a = b = 3.2535(6) Å and c =5.2142(8) Å for ZA2 sample. The lattice constants have the same variations as in the case of Z1 and ZA1 samples. The ZnO crystal is almost strain-free in Z2 and ZA2 samples (0.07 and 0.09, respectively). By comparison, in the XRD pattern of the Z2 sample, some other peaks appeared in addition to those observed in the Z1 sample (Figure 2), corresponding to the SDBS crystals diffraction patterns, which were present due to the limited solubility of SDBS in ethanol. Due to residual L-lysine, more diffraction peaks were present in the X-Ray pattern of the ZA2 sample. 

The resulting nanomaterials have high crystallinity since all peaks are sharp, as seen in Figure 2, and no further annealing treatment was needed. The crystallite size was calculated based on the full width at half-maximum of the (1 0 1) diffraction peak using Debye–Scherrer formula and resulted in a value of 11.9 nm for Z1 compared to 10.4 nm for those decorated with Au-ZA1 indicating a slight decrease in crystallinity after Au NPs synthesis step. When only ethanol was used as a solvent in syntheses, the crystallite size is increased at 15.8 nm for Z2 and 15.1 nm for ZA2.

The morphology and microstructure of the ZnO and ZnO/Au samples were observed with SEM, and the images are shown in Figure 3a–d. ImageJ software was used to measure the individual dimensions of the ZnO nanorods and the Au nanoparticles from the recorded SEM micrographs. The histogram distributions of ZnO nanorod lengths are presented in Figure 3e,f, and the diameter of Au nanoparticles in the ZA1 and the ZA2 nanocomposites is in Figure 3g,h. Histograms of ZnO nanorods diameter distribution in samples Z1 and Z2 are presented in Appendix A. 

The scanning electron microscope image of the Z1 sample (Figure 3a) shows ZnO nanorods with very small dimensions compared to the dimensions reported in the literature for ZnO nanorods [47,50]. The nanorods have a relatively narrow dispersion of dimensions between 20 and 78 nm. The mean length of nanorods was 55.0 ± 1.3 nm (Figure 3e). The diameters were 6–20 nm, with an average of 11.8 ± 0.2 nm (Appendix A), and the aspect ratio was more than 4. When samples were obtained through the solvothermal method using only ethanol as solvent, the SEM image revealed the formation of ZnO nanorods with a less regular shape and more agglomerated (Figure 3b). The size of these nanorods varied from 22 nm to 98 nm in length, predominating those of 69.5 ± 2.1 nm (Figure 3f), and the diameters were in the 12–36 nm range, with a mean value of 18.5 ± 0.6 nm (Appendix A), with an aspect ratio less than 4. In addition, some long lamellar structures stood out from the SEM images, probably due to SDBS crystallization during synthesis. 

SEM images also showed that the growth of ZnO crystals in ethanol/water mixture was different from that in ethanol only. Furthermore, the sodium dodecyl benzene sulfonate acted as a structure-directing growth agent. It was used to obtain tiny nanorods through anisotropic growth and to prevent the aggregation of the formed nanostructures [51], and this effect is more accentuated during the solvothermal synthesis route when a small quantity of water is present (Z1 sample). When only ethanol was used as a solvent, zinc oxide nanostructures’ growth was less directed by the SDBS (Z2 sample), and these differences in dimensions and morphology between the two products could be related to decreased solubility of SDBS in ethanol compared to water. SDBS ionizes entirely in water, and the resulting anion can be adsorbed onto certain crystal facets, leading to anisotropic growth of the particles. In the solvothermal synthesis of ZnO, water and ethanol can act as a solvent and reactants [52]. The differences in polarity and even in kinetic diameter of the two types of substances influenced the different stages of zinc oxide particle formation and even the possible aggregation of small particles [53]. In addition, the difference in saturated vapor pressures of the two reaction environments leads to dissimilar nanostructures’ growth. Z1 nanorods resulted from a solvent mixture with a lower vapor pressure than in the Z2 case, which should allow for the more rapid growth of the ZnO nuclei, resulting in larger particles [54], but the SDBS is more accessible in the Z1 case to control the crystal growth. The two factors, the solubility of the reactants and the vapour pressure of the solvents, acted simultaneously, controlling the morphology of the resulting nanostructures. Moreover, the tendency to form agglomerations of NPs is more accentuated for solvents with higher saturated vapor pressure (the Z2 case). 

In the case of ZnO nanorods modified with Au NPs, the SEM image (Figure 3c) showed the formation of small NPs in addition to ZnO nanorods in sample ZA1 having diameters from 4 nm to 11 nm and a mean value of 8.5 ±0.12 nm (Figure 3g). After the second step of the synthesis, small spherical Au NPs also appeared in sample ZA2, surrounded by a lysine cloud (Figure 3d). These gold nanoparticles are characterized by a mean value of 6.2 ± 0.2 nm in diameter (Figure 3h). 

The morphological feature of synthesized samples was further analyzed with TEM. ZnO nanorods are well crystallized, as observed from the TEM analysis of the ZA1 (Figure 3i) sample and ZA2 sample (Figure 3j). As seen in the enlarged TEM image of the ZA2 sample, the ZnO nanorods have a more irregular contour (Figure 3k). The formation of some agglomerates was observed, probably due to the presence of the functional groups—hydroxyl, sulphate, carboxyl and amines—on NPs surface, which can cause the formation of weak van der Waals forces and hydrogen bonds between the nanostructures. 

The compositional analysis conducted through the energy dispersive X-ray technique proved the Au presence in the ZnO/Au samples (Figure 3l). The EDX spectrum of the ZA2 sample shows a high signal for zinc (Zn Lα) and oxygen (O Kα), and an absorption band at approximately 2.2 keV, assigned to binding energy for Au nanocrystallites [55]. Some other weak bands are also present, corresponding to the trace amounts of S and K that remained from synthesis. EDX elemental mapping of ZA2 nanocomposite (Appendix A) shows a uniform distribution of elements in the composite.

ATR-FTIR spectroscopy detected the presence of different functional groups as a result of the synthesis process. The FTIR spectra, compared in Figure 4, showed broad bands at 374 and 382 cm^−1^ for Z1 and ZA1 samples, respectively, and at 391 and 412 cm^−1^ for Z2 and ZA2 samples, respectively, corresponding to the metal-oxygen (Zn-O) vibration modes. The results confirmed the successful formation of ZnO with a wurtzite structure. The difference in the shape and position of the bands can be related to a change in the lattice constants and the morphology of ZnO nanostructures [56,57], as demonstrated by XRD and SEM images. The band position of zinc-oxygen vibration in the lattice is in accordance with the values published in the literature [58]. The FTIR spectrum of the Z1 sample showed no other high-intensity transmission bands, demonstrating the successful removal of organic residues from the precipitate (also proved by the results of the XRD analysis). Other bands appeared in the FTIR spectra of the ZA1, Z2 and ZA2 samples, corresponding to the stretching mode of the SO (1374 cm^−1^) bond from SDBS and the bending mode of CH (2950–2855, 1500–1200 cm^−1^) bonds from SDBS chain and L-lysine, respectively. The FTIR bands attributed to SDBS traces can be observed slightly shifted to higher wavenumbers, compared to the raw material, probably due to the formation of hydrogen bonds between the -SO_3_^−^group and H_2_O or ethanol molecules that may lower the energy in the system [59]. The bands were more intense for the Z2 and ZA2 samples than for the Z1 and ZA1 due to the residual quantity of SDBS after synthesis. The widened bands from 1439 cm^−1^ of ZA1 and ZA2 samples correspond to the O-H bending of the carboxylic group from lysine that may be involved in forming hydrogen bonds. The absorption bands confirmed the interaction of the carboxylic group (1760 cm^−1^, which becomes smaller) from lysine with the –OH groups on the surface of the Au NPs. The band at 1183 cm^−1^ was assigned to the rocking deformation of NH_3_^+^. The absorption band centered at approximately 1100 cm^−1^ can be associated with the symmetrical vibration of the C-O bonds from lysine. The results confirmed the presence of an L-Lysine capping agent on the Au NPs. In the 1000–600 cm^−1^ domain, few bands were recorded, and they can be assigned to the vibrations of the backbone bonds from lysine or SDBS, peaks centered at 989 (C-C-H) 842 (C-N), 822 (C-OH), and 617 cm^−1^ (C-H). 

Size distribution profiles and colloidal stability of all samples were determined by DLS. Suspensions of each sample in distilled water (0.1 mg/mL) were obtained as in [60] using an ultrasonic probe fitted with a vial tweeter. The optimized parameters for sonication were found to be 10 W with amplitude adjustment at 90 % for 2 min. The hydrodynamic diameter values (Appendix A) are placed in two regions due to the ZnO non-spherical geometry of ZnO nanorods (Table 1) and are larger than the dimensions given by the SEM and TEM analyses. In the ZA1 and ZA2 samples, the population of smaller particles increased due to the presence of supplementary small L-lysine-capped Au NPs. The extent of dynamic diameters was influenced by the SDBS or α-amino acid (L-lysine) on NPs’ surfaces (confirmed by FTIR data), indicating a reasonably thick coating around NPs and interactions between the functional groups of these coatings. Polydispersity index (P.D.I.) values for these multiple peaks distributions of hydrodynamic diameters vary between 0.446 (sample Z1) and 0.589 (sample ZA2).

Zeta potential measurements showed that the samples Z1 and Z2 have a positive charge and ZA1 and ZA2 have a resulting small net negative charge. Considering the generally accepted condition for stable colloids, |ζ| ≥ 15 mV [61], the most stable suspension is Z1, with a potential of 17.77 mV. The ζ-potential of the Z2 sample (14.09 mV) facilitates the agglomerates visualized in the SEM analysis. Lysine-capped Au NPs had a high negative charge [62,63] in contact with ZnO NPs, which led to a change in the electrical charge of the composite system. In addition, the interactions between functional groups on the surfaces of the ZnO and Au NPs may be responsible for the increased dynamic size of NPs.

### 3.2. Antibacterial Activity 

We used some of the usual methods for testing the antimicrobial activity of nanoparticles: one qualitative (disk diffusion assay) and one quantitative (tetrazolium/formazan assay). The TTC test is considered reliable [64]. In addition, as in the case of any analysis method, some problems may arise when titration-inoculation is used. For example, both Escherichia coli and Staphylococcus aureus produced aggregates in liquid batch cultures, reported as in [65].

Figure 5a–c show the results of quantitative/qualitative evaluation of the antibacterial activity of ZnO and ZnO/Au nanomaterials using *Escherichia coli* (Gram-negative) and *Staphylococcus aureus* (Gram-positive) bacterial strains. The qualitative antibacterial assay employed the standard Kirby–Bauer disk diffusion method and the diameters of inhibition zones are tabulated in Table 2. This assay remains one of the most widely used antimicrobial susceptibility testing methods. It is frequently applied to study the antimicrobial performance of nanoparticles with some adaptations of the classical method. Even if other methods for antimicrobial susceptibility testing were developed, disk-diffusion assay offers many advantages: basic equipment, extensive applicability, and straightforward interpretation. The quantitative evaluation of bacterial viability against Gram-negative bacteria, *E. coli* and Gram-positive bacteria, *S. aureus*, is shown in Figure 5c. 

A one-way ANOVA was conducted to compare the effect of the samples’ antibacterial activity, and we found a statistically significant result. Table 2 shows that the calculated value of the Fisher statistical test (F) is higher than the critical value (F crit), and the value associated with the probability (*p*-value = 0.005) is lower than the risk value (α value = 0.05). This result leads us to reject the null hypothesis, according to which the model is not statistically significant. At least two means differ highly significantly. Therefore, with a probability of 95%, the model is statistically significant: the dependent variable, bacteria (*E. coli*), is significantly influenced by the variation of the independent variable, Au NPs modified ZnO nanorods (ZA1), comparable with unmodified ZnO (Z1), respectively, (ZA2) comparable with (Z2). The same statement is also applicable in the case of *S. aureus*. The microbiological experiments showed that gold addition to ZnO nanomaterials increased the overall antibacterial performance against both bacteria, with a more pronounced effect in the case of the ZA1 sample. The values for the ZI are higher in the *E. coli* bacteria test than in the *S. aureus* test. The same trend is shown by the results obtained from the TTC test, which measures the oxidoreductase activity of the cells through photocorrosive reactions in aqueous media [66]. The reduction in cell viability was approximately 83% for ZA1, 81 % for ZA2, 58 % for Z2 and 57 % for Z1 in the case of *E. coli*, while for *S. aureus,* it was 82 % for ZA1, 76 % for ZA2, 43 % for Z2 and 42 % for Z1. One-way ANOVA applied to TTC test results proved very strong evidence against null hypothesis (*p* < 0.001).

The bactericidal mechanism of ZnO and Au, such as in the case of all NPs, is complex. NPs can interact with living bacterial cells both physically and chemically. Some of these processes can coincide, contributing to overall antibacterial activity. In an aqueous medium, ZnO gradually releases ions due to the chemical potential in the solution [67]. Zn^2+^ ions can enter more easily in more negatively charged *E. coli* than in *S. aureus* bacteria due to electrostatic interactions, leading to increased uptake of ions. Once inside, Zn^2+^ can cause cell structural changes and aberrant enzyme activities. The difference in activity against the two types of bacteria may be attributed to the differences in structure and chemical composition of the cell wall. *S. aureus* has a cytoplasmic membrane and thick cell walls composed of several layers of peptidoglycan. In contrast, *E. coli* has a more complex cell wall structure, with layers of peptidoglycan between the outer membrane (lipopolysaccharide molecules with negative charges) and the cytoplasmic membrane. This difference in the cell wall structure and its electrical charge can facilitate the entrance of positive ions released by the ZnO into the *E. coli* cell, which explains the better antibacterial results recorded against these bacteria strains (Figure 5). 

When ZnO or Au nanostructures and bacteria cells are in close contact with each other, direct physical interactions may occur that can lead to surface electric charge neutralization, membrane deformation and deterioration and changes in its penetrability, causing even its rupture and leakage of cytoplasmic contents out of the cell leading to bacterial death [68]. These electrical interactions between the two types of antibacterial materials differ as ZnO are positively charged, and ZnO/Au ones are slightly negatively charged (see Table 1). In this proposed mechanism, *Staphylococcus aureus*, a Gram-positive bacterium, proved to be more protected due to the thick peptidoglycan layer (up to 40 layers), which confers significant mechanical strength. Surface modification of Au NPs with L-lysine ensures a better dispersion of the ZA1 and ZA2 nanocomposites in the water but also contributes to the overall antimicrobial effect through strong ionic interaction with the bacteria. Au NPs surface modification by different coating agents has proved to induce distinct changes in the antibacterial effects [40]. L-lysine as cationic amino acid interacts preferentially with microbial membranes using electrostatic interaction between their cationic charge and anionic charge on the membrane surface. Recently, it was demonstrated that 100 nm Au NPs can kill bacteria through mechanical deformation of the cell wall when they come in contact with the cells [69]. These nanomaterials’ morphological and structural characteristics are essential in their interactions with the studied bacteria. By comparison, the Z1 sample showed better antibacterial performance than Z2, meaning that Z1, with a large interfacial area (high surface area per unit of volume), enters the membrane easier, resulting in superior antibacterial efficiency. Other researchers also found those smaller particles (agglomerates) have a much better antibacterial activity [68,70]. Furthermore, Z1 nanorods have a higher aspect ratio than Z2 nanorods, with a superior fraction of unsaturated Zn^2+^ sites meaning more defects due to oxygen vacancies [71], resulting in higher reactivity against bacterial cells. Au NPs interact with both types of bacteria due to their very small size, which allows for better interactions with all kinds of bacteria. In addition, the difference in morphology between these nanomaterials is reflected in the antimicrobial activity, sample Z1 being more reactive due to plentiful corners and edges compared with the Z2 sample [72]. 

An alternative antimicrobial mechanism of ZnO and Au NPs could be through reactive oxygen species (ROS), such as hydrogen peroxide or superoxide anions [73,74,75]. Aquatic ZnO and ZnO/Au dispersions produced an augmented level of ROS, promoting severe oxidative stress and destroying cellular components (lipids, enzymes, DNA, and proteins) [76]. Au NPs with small dimensions proved to induce the formation of holes in the membrane of *E. coli* through the ROS mechanism [77] and increase of concentration of intracellular ROS species in *S. aureus* after internalization [78]. The high surface area per unit of volume in the case of all obtained nanostructures generated more reactive oxygen species. In addition, ZnO crystal has rich defect chemistry [79], which may contribute to its increased antimicrobial activity. Experimental determination of one of the ROS (H_2_O_2_) species generated by our samples in dark conditions was conducted through redox titration. However, this method failed to determine the appearance of this reactive species. One possible explanation would be the unsuitability of this method for our materials. The qualitative results of hydrogen peroxide generation from ZnO and ZnO/Au suspensions are presented in Appendix A.

In addition, when the test bacteria were put in contact with the ZnO and ZnO/Au nanomaterials, the bacterial cells produce excessive ROS, which can no longer be counterbalanced by enzymes and antioxidants, leading to bacteria damage [80]. The reactive oxygen species secreted by the bacteria lead to the reduction of the 2,3,5,-triphenyl tetrazolium chloride in TTC test. Furthermore, the TTC results showed a reduction in cell viability in ZA1 and ZA2 samples due to Au nanoparticles presence.

Comparing the antimicrobial performance of our ZnO nanomaterials can be difficult since antimicrobial efficacy is measured by different methods (e.g., minimum inhibitory concentration, zone of inhibition). This efficacy is linked to the morphology and structure of the nanomaterials used and bacterial colonies’ features. Our results are comparable with those obtained by other nanomaterials having similar sizes, e.g., Ag NPs/ZnO nanohybrid [81] or ZnO nanoflowers [82] or ZnO NPs [25], Ag doped ZnO NPs and Au doped ZnO NPs [83].

Kaushik et al. studied the antimicrobial activity of nanoparticles with diameters between 82 and 420 nm, and observed that the zone of inhibition for *E.coli* (ATCC35150) and *S. aureus* (ATCC 13150) decreases with increasing particle sizes [25]. The results were slightly better for *S. aureus* (from 29 to 17 mm), compared to *E. coli* (from 25 to 17 nm), the differences being more pronounced for smaller nanoparticles. The concentration of nanoparticles used in the antimicrobial tests was not reported. ZnO NPs having 20–30 nm sizes were introduced in bacterial cellulose with a concentration of 1.68 µg ZnO NPs/mm^2^, and proved to prevent completely the adherence of *E. coli* cells onto substrate [15]. Gunalan et al. [24] proved that green ZnO NPs (40 nm sizes) show more enhanced biocidal activity against *S. aureus* than chemical ZnO NPs (25 nm sizes), and the maximum activity was 26 mm ZI at 6 mM concentration of ZnO NPs. Other 100 nm ZnO nanoparticles exhibited large inhibition zones for Escherichia coli (16 mm), and Staphylococcus aureus (12 mm) using a 30 µg mL^−1^ sample concentration [84]. ZnO/Au (1%) nanocomposites made of ZnO NPs (obtained using the hydrothermal method) and Au NPs (synthesized through chemical reduction) were tested against *S. aureus* and *E.coli* [46]. ZnO NPs had 19.8 nm mean diameter and the partially spherical Au NPs had 20–40 nm dimensions. The ZI were found to be 9.0 mm for *S. aureus* and 13.4 mm for *E. coli*, very close to the antibacterial effect caused by ZnO NPs alone.

We also obtained Au nanoparticles (NP) in the absence of ZnO nanorods; however, these AuNPs have larger dimensions than those from the composite, as demonstrated by the change in color of the resulted colloidal solution. This observation means that the ZnO NR’s presence influenced the morphology of Au NPs, as the germination and growth of AuNPs happened differently. Due to this difference in morphology and the small amount of gold (0.2% wt) in the composite, we considered not presenting here these results.

Although they have a weaker efficiency against microbial strains than Ag NPs [84], ZnO and ZnO/Au nanomaterials have some advantages, which may support their further potential. For instance, the ZnO has a low cost and a white appearance, and Au NPs are considered highly biocompatible [85], stable under visible light and require less constraining synthesis conditions. Coatings made of Au and ZnO nanomaterials can withstand high-temperature sterilization when conventional antibiotics are inactivated. Our nanomaterials could be used for antimicrobial coatings to inhibit bacterial adhesion and biofilm formation.

## 4. Conclusions

The present study was focused on the wet chemical-based synthesis (solvothermal and chemical reduction in solution) of zinc oxide nanostructures. ZnO nanorods with different aspect ratios were obtained using SDBS as a structure-directing agent. L-Lysine capped Au NPs were in-situ synthesized and supported on ZnO nanorods. The XRD, SEM and TEM-based analysis showed the ZnO and ZnO/Au crystalline structures and morphologies. FTIR spectroscopy analysis proved the existence of new bonds in the obtained nanomaterials, and DLS analysis determined the NPs’ surface charge and stability in solutions. The ZnO and ZnO/Au powders exhibited efficient antibacterial activity against *E. coli* and *S. aureus* strains. The ZnO nanorods with smaller sizes have higher antibacterial activity against bacteria than bigger ZnO nanorods. The association of ZnO in nanorods and lysine-coated Au NPs increased the antimicrobial activity against the Gram-positive and Gram-negative bacteria. Therefore, the ZnO/Au nanomaterials are potential antimicrobial agents. Since the advantages such as biocompatibility, high-temperature resistance, and cost-effective synthesis, further consolidate the ZnO and ZnO/Au NPs profile and potential, future studies, in vitro and in vivo, will conclude the best formulations for successful implementation in the healthcare industry for the maximum benefit of the patients, professionals, and society.

## Figures and Tables

**Figure 1 nanomaterials-12-03832-f001:**
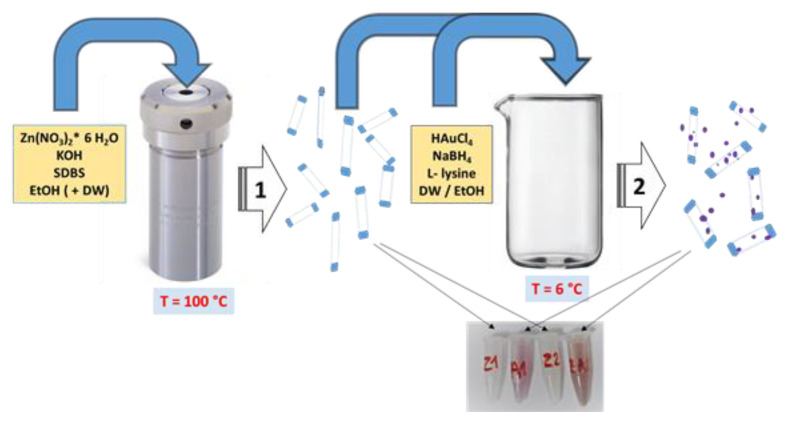
Flow chart of of ZnO and ZnO/Au synthesis.

**Figure 2 nanomaterials-12-03832-f002:**
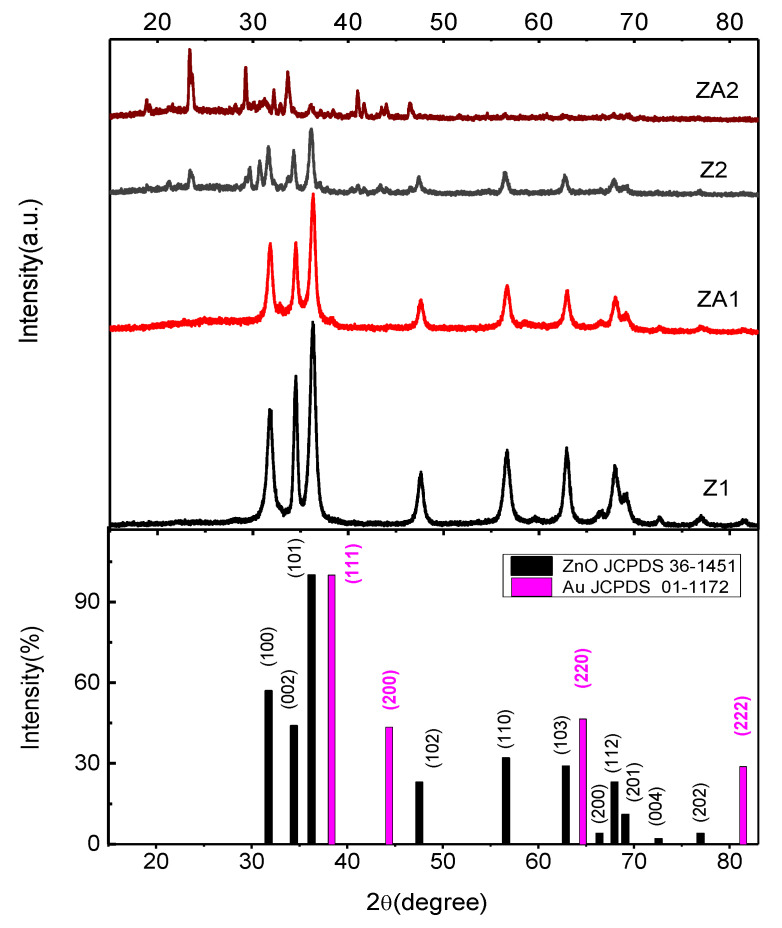
XRD patterns of ZnO and ZnO/Au nanomaterials.

**Figure 3 nanomaterials-12-03832-f003:**
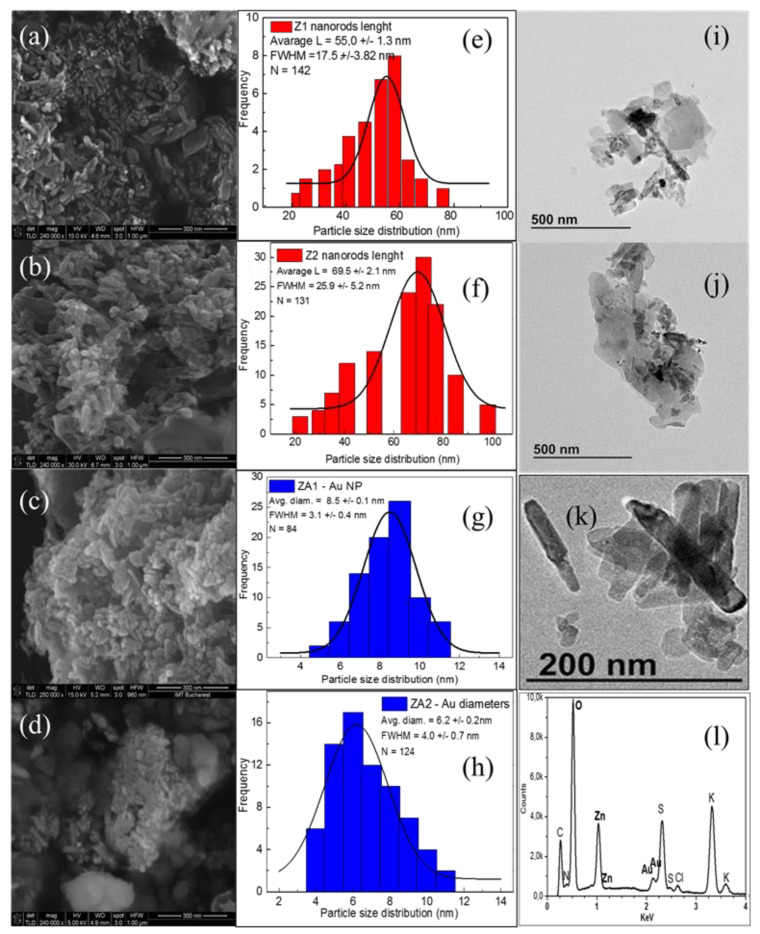
SEM images of the (**a**) Z1, (**b**) Z2, (**c**) ZA1, and (**d**) ZA2 samples; Histogram distributions of the (**e**) Nanorods length in Z1, (**f**) Nanorods length in Z2, (**g**) Au NPs diameters in ZA1, and (**h**) Au NPs diameters in ZA2, (**i**) TEM image of ZA1, (**j**) TEM analysis of ZA2; (**k**) TEM magnification of sample ZA2 (**l**) EDX analysis of ZA2 sample.

**Figure 4 nanomaterials-12-03832-f004:**
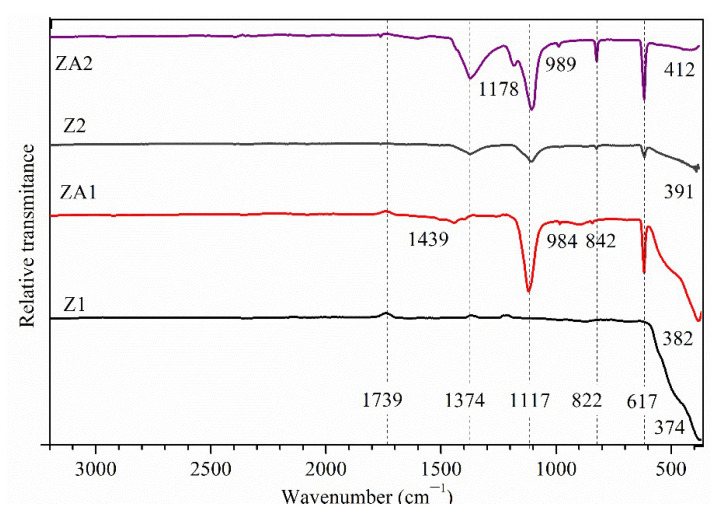
ATR-FTIR spectra for ZnO (Z1 and Z2) and ZnO/Au (ZA1 and ZA2) samples.

**Figure 5 nanomaterials-12-03832-f005:**
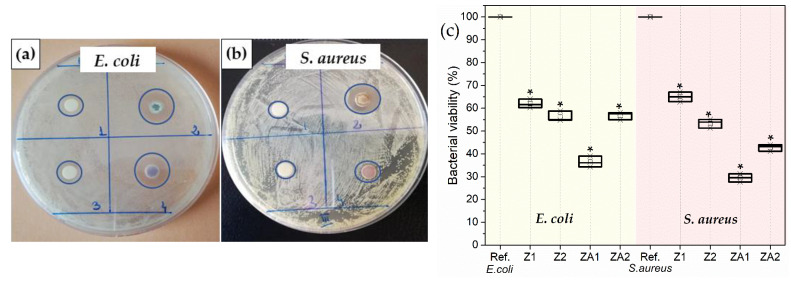
Disc diffusion assay- inhibition zone formed against the growth of (**a**) *E. coli* and (**b**) *S. aureus*: 1- Z1, 2- ZA1, 3-Z2, 4-ZA2; (**c**) Bacterial viability by TTC assay for *E. coli* and *S. aureus*. The data represent the mean ± SD of three replicates. * denotes the statistical significant difference compared with controls (*p* < 0.001).

**Table 1 nanomaterials-12-03832-t001:** DLS measurements of ZnO and ZnO/Au nanomaterials.

Sample	Population 1		Population 2		P.D.I.	ζ/mV
	%	d_h_/nm	%	d_h_/nm		
Z1	39	23	61	100	0.446	17.77 ± 1.41
ZA1	47	23	53	105	0.493	−4.34 ± 0.80
Z2	31	33	69	115	0.580	14.09 ± 1.24
ZA2	35	33	65	121	0.589	−5.18 ± 0.91

**Table 2 nanomaterials-12-03832-t002:** The diameters of the inhibition zones for *E. coli* and *S. aureus* strains.

*Samples*	Z1		Z2	ZA1		ZA2
*ZI (mm)/E. coli*	8		7	14		12
*ZI(mm)/S.aureus*	6		6	14		10
*Sum*	14		13	28		22
*Average*	7		6.5	14		11
*Variance*	2		0.5	0		2
*Source of Variation*	*SS*	*df*	*MS*	*F*	*p*-*Value*	*F crit*
Between Groups	75.375	3	25.125	22.3333	0.0058	6.5913
Within Groups	4.5	4	1.125			
Total	79.875	7				

## Data Availability

Not applicable.

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
