# Peer review of "Synthesis of ZnO/Au Nanocomposite for Antibacterial Applications"

_nanomaterials, 2022, doi:10.3390/nano12213832_

Round 1
Reviewer 1 Report (Previous Reviewer 2)
The resubmitted manuscript is much better than the former version. However, some questions still need to be answered. Minor revision is suggested.
1. “aAu/ZnO” should be revised as “Au/ZnO” for keywords.
2. The authors should pay attention to the writing of units. There should be a space between the number and the unit.
3. How about the antibacterial activity of Au nanoparticles to E. coli and S. aureus?
4. How about the influence of dosage of Au/ZnO on the antibacterial activity?
5. Please pay attention to the references. Ref.10 is missing. The writing of journal name should be in the same style. Page number and volume number are missing for some references. Please pay attention to the writing of subscripts and superscripts of compounds in the references.
Author Response
Response to the reviewer’ comments:
- “aAu/ZnO” should be revised as “Au/ZnO” for keywords.
Response: This keyword was modified.
- The authors should pay attention to the writing of units. There should be a space between the number and the unit.
Response: The writing of units was revised, and the manuscript has been updated accordingly.
- How about the antibacterial activity of Au nanoparticles to E. coli and S. aureus?
Response: We also obtained Au nanoparticles (NP) in the absence of ZnO nanorods; however, these AuNPs have larger dimensions than those from the composite, as demonstrated by the change in colour of the resulted colloidal solution. This observation means that the ZnO NR’s presence influenced the morphology of Au NPs, as the germination and growth of AuNPs happened differently. Due to this difference in morphology and the small amount of gold (0.2% wt) in the composite, we considered not presenting these results in this article, as it can be the subject of another research. (Line 486-492)
- How about the influence of dosage of Au/ZnO on the antibacterial activity?
Response: For a fair comparison, our study used a concentration value within the range of concentrations applied to test of antibiotics in antibiograms. (Line 193-194)
- Please pay attention to the references. Ref.10 is missing. The writing of journal name should be in the same style. Page number and volume number are missing for some references. Please pay attention to the writing of subscripts and superscripts of compounds in the references.
Response: Reference 10 was corrected. We checked and updated when required. Page number and volume number were missing from some references due to the format of the respective journals.
Reviewer 2 Report (New Reviewer)
This work is devoted to solving a global problem - the search for new methods for the fight against bacteria that have high resistance. The presence of antibacterial properties at nanoparticles has long been known. There are many studies on the influence of shape, size, etc. for antibacterial properties. In this work, ZnO and ZnO/Au nanoparticles were proposed, which were synthesized by the solvothermal method. A wide range of methods have been used to study nanoparticles. Gram-positive (Staphylococcus aureus) and Gram-negative bacteria (E. coli) were selected for research. There are used two methods of disk diffusion and tetrazolium/formazan (TTS) analysis.
Biofilms are of the greatest interest. As far as I understand, plankton culture was used in the work?
Why is the microbiological method of titration-inoculation not used? He is reliable.
- I consider it important to compare the results of antibacterial tests obtained for your nanoparticles with others and authors studying ZnO nanoparticles. Including when using other methods of obtaining.
- In Fig. 3 there is a typo in the signature (not b but k)
- In the text, where with a capital letter Gram is positive, where with a small one. Correct with a large one, since the method is named after Gram (p. 12).
- Are there any in-vivo experiments planned?
- References should be made according to the rules of the journal.
- Reference 10-?
Author Response
Response to the reviewer’ comments:
Biofilms are of the greatest interest. As far as I understand, plankton culture was used in the work?
Response: Indeed, biofilm is a topic of interest, and we can use our nanomaterials for antimicrobial coatings to inhibit bacterial adhesion and biofilm formation. We used Escherichia coli and Staphylococcus aureus as test bacteria. We streaked on an agar culture medium from the stock culture and obtained colonies. Then, a colony was inoculated in a liquid medium to carry out the tests. (Line 499-500)
Why is the microbiological method of titration-inoculation not used? He is reliable.
Response: We used some of the usual methods for testing the antimicrobial activity of nanoparticles: one qualitative (disk diffusion assay) and one quantitative (tetrazolium/formazan assay). The TTC test is considered reliable [Braissant, O., Astasov-Frauenhoffer, M., Waltimo, T., & Bonkat, G. (2020). A review of methods to determine viability, vitality, and metabolic rates in microbiology. Frontiers in Microbiology, 11, 547458]. Also, as in the case of any analysis method, some problems may arise when titration-inoculation is used. For example, both Escherichia coli and Staphylococcus aureus produced aggregates in liquid batch cultures, reported as in [Kragh, K. N., Alhede, M., Rybtke, M., Stavnsberg, C., Jensen, P. Ø., Tolker-Nielsen, T., ... & Bjarnsholt, T. (2018). The inoculation method could impact the outcome of microbiological experiments. Applied and environmental microbiology, 84(5), e02264-17]. (Line 370-380)
- I consider it important to compare the results of antibacterial tests obtained for your nanoparticles with others and authors studying ZnO nanoparticles. Including when using other methods of obtaining.
Response: The antimicrobial performances of our nanomaterials were compared with those of other ZnO-based nanomaterials with similar dimensions, obtained by different methods, and selected materials that were tested using the same methods as those presented in our article.
“Comparing the antimicrobial performance of our ZnO nanomaterials can be difficult since antimicrobial efficacy is measured by different methods (e.g., minimum inhibitory concentration, zone of inhibition). This efficacy is linked to the morphology and structure of the nanomaterials used and bacterial colonies’ features. Our results are comparable with those obtained by other nanomaterials having similar sizes, e.g., Ag NPs/ZnO nanohybrid [76] or ZnO nanoflowers [77] or ZnO NPs [23], Ag doped ZnO NPs and Au doped ZnO NPs [78].” (Line 480-485)
- In Fig. 3 there is a typo in the signature (not b but k)
Response: The correction was made in the Fig. 3 caption.
- In the text, where with a capital letter Gram is positive, where with a small one. Correct with a large one, since the method is named after Gram (p. 12).
Response: The type of bacteria were rewritten so that the first letter is capitalized.
- Are there any in-vivo experiments planned?
Response: Currently, our research facilities do not include an accredited laboratory with environmental conditions for conducting the in-vivo tests. For further research development, we propose identifying collaborators with such laboratories for a future research proposal. (Line 516)
- References should be made according to the rules of the journal.
Response: The references were revised according to the rules of the journal. Still, some references cannot be formulated according to the rules of this journal, e.g., some of them have no page range.
- Reference 10-?
Response: Reference 10 was introduced in the correct form.
Reviewer 3 Report (New Reviewer)
My comments are in attachement.

Author Response
Response to the Reviewer 3 comments:
Academy of Romanian Scientists, 010071 Bucharest,
What is the country?
Response: The country is Romania, we updated the affiliation.
Annually antimicrobial resistance causes about 700,000 deaths worldwide and is an accelerated growth trend due to the increased use of antibiotics in the coronavirus pandemic.
Annually, weak antimicrobial resistance causes about 700,000 deaths worldwide and is an accelerated growth trend due to the increased use of antibiotics in the coronavirus pandemic.
Response: we rephrased (Line 18-20) and introduced more data (Line 47-53).
Annually, antimicrobial-resistant infections-related mortality worldwide accelerates due to the increased use of antibiotics during the coronavirus pandemic and the antimicrobial resistance, which grows exponentially, and disproportionately to the current rate of development of new antibiotics. (Line 18-20)
Page 1
The motivation behind this work was to find a superior antibacterial nanomaterial, which can be efficient, biocompatible, and stable over time.
The motivation behind this work was to find a superior antibacterial nanomaterial, which can be efficient, biocompatible, and stable in time.
Page 2
For instance, the metal oxide nanoparticles (NPs) intensively researched over the last two decades proved to be a new line of defence against multi-drug resistance microorganisms.
For instance, the metal oxide nanoparticles (NPs) intensely investigated over the last two decades, and they proved to be a new line of defence against multi-drug resistance microorganisms.
Page 2
Although their activity is slower than the organic antibacterial agents, the NPs do not release dangerous by-products [9].
Although their activity is slower than the organic antibacterial agents, NPs do not release dangerous by-products [9].
Page 2
Although ZnO and Au NPs have been tested independently against different bacteria, a few papers have reported using combinations of the two NPs against bacterial pathogens: Au decorated ZnO NPs [28, 43].
Although ZnO and Au NPs were tested independently against different bacteria, a few contributions reported using combinations of the two NPs against bacterial pathogens: Au decorated ZnO NPs [28, 43].
Page 3
ZnO nanostructures were obtained by solvothermal method without a seeding step, starting from a modified method published in reference [44].
ZnO nanostructures were obtained by solvothermal method without a seeding step, starting from a modified method reported in [44].
Page 3
Au NPs were obtained through the Brust-Schiffrin method, starting from chloroauric acid using sodium borohydride as a reducing agent.
Au NPs were obtained through the Brust-Schiffrin method, starting from chloroauric acid, using sodium borohydride as a reducing agent.
Page 3
In a Berzelius beaker containing 30 ml of ethanol and 10 ml of deionised water, 0.8925 g of zinc nitrate hexahydrate was dissolved by stirring for 20 minutes.
In a Berzelius beaker containing 30 ml of ethanol and 10 ml of deionised water, 0.8925 g of zinc nitrate hexahydrate was dissolved by stirring for 20 min.
Page 3
The solution was stirred for 30 min, transferred to a 45 ml Teflon-lined stainless-steel pressure vessel (autoclave) and placed in the preheated oven at 100 °C for 10 hours.
The solution was stirred for 30 min, transferred to a 45 ml Teflon-lined stainless-steel pressure vessel (autoclave) and placed in the preheated oven at 100 °C for 10 h.
Page 3
Then the obtained powder was washed several times with ethanol and deionised water. The resulting product was oven-dried at 60 °C for 12 hours
Then, the obtained powder was washed several times with ethanol and deionised water. The resulting product was oven-dried at 60 °C for 12 h.
Page 4
0.1 mM sodium borohydride aqueous solution kept at 6°C in 0.1 ml portions was added until the colour changed from white to faint purple, and the magnetic stirring continued for 30 minutes. The resulting suspension was left for 16 hours and then centrifuged. The residue was washed repeatedly with ethanol and water and dried in an oven at 60 °C for 8 hours.
0.1 mM sodium borohydride aqueous solution kept at 6°C in 0.1 ml portions was added until the colour changed from white to faint purple, and the magnetic stirring continued for 30 min. The resulting suspension was left for 16 h and then centrifuged. The residue was washed repeatedly with ethanol and water and dried in an oven at 60 °C for 8 h.
Page 4
The concentrations of Zn and Au in synthesised nanomaterials were determined using inductively coupled plasma optical emission spectrometry (ICP-OES) Optima 5300 DV Perkin Elmer.
The concentrations of Zn and Au in synthesised nanomaterials were determined by inductively coupled plasma optical emission spectrometry (ICP-OES) using an Optima 5300 DV Perkin Elmer.
Page 4
The bacteria strains were grown at 37°C for 18 hours to obtain the working cultures.
The bacteria strains were grown at 37°C for 18 h to obtain the working cultures.
Page 4
Antimicrobial susceptibility testing discs (Bioanalyse) were impregnated with 10 μL of ZnO or ZnO/Au suspensions, allowed to dry for 10 minutes and placed in triplicate on the seeded MHA plates. The testing plates were incubated at 37°C for 12 hours, and the inhibition zones’ diameters were measured.
Antimicrobial susceptibility testing discs (Bioanalyse) were impregnated with 10 μL of ZnO or ZnO/Au suspensions, allowed to dry for 10 min and placed in triplicate on the seeded MHA plates. The testing plates were incubated at 37°C for 12 h, and the inhibition zone diameters were measured.
Page 4
The flasks were incubated while agitating (200 rpm) for 3 hours at 37 °C. 1 ml from the control sample, and experimental samples were added into sterilised testing tubes containing 100 μL TTC (0.5%, w/v), then incubated at 37 °C for 20 minutes.
The flasks were incubated while agitating (200 rpm) for 3 hours at 37 °C. 1 ml from the control sample, and experimental samples were added into sterilised testing tubes containing 100 μL TTC (0.5%, w/v), then incubated at 37 °C for 20 min.
Page 5
Comparing the XRD patterns of Au/ZnO with those of the ZnO sample in Fig. 2 showed they are similar, meaning that the formation of Au during the reduction reaction did not significantly influence the crystal structure of ZnO.
Comparing the XRD pattern of Au/ZnO with that of the ZnO sample in Fig. 2 showed they to be similar, meaning that the formation of Au during the reduction reaction did not noticeablly influence the crystal structure of ZnO.
Page 6
In the case of samples obtained using only ethanol as a solvent, the wurtzite ZnO lattice parameters are a = b = 3.2539(6) Å and c =5.2161(10) Å for the Z2 sample and a = b = 3.2535(6) Å and c =5.2142(8) Å for ZA2 sample. The lattice parameter values have the same variations as in the case of Z1 and ZA1 samples.
In the case of samples obtained using only ethanol as a solvent, the wurtzite ZnO lattice constants are a = b = 3.2539(6) Å and c =5.2161(10) Å for the Z2 sample and a = b = 3.2535(6) Å and c =5.2142(8) Å for the ZA2 sample. The lattice constants have the same variations as in the case of Z1 and ZA1 samples.
Page 6
Due to residual L-lysine, more diffraction peaks were present in the X-Ray diffractogram of the ZA2 sample.
Due to residual L-lysine, more diffraction peaks were present in the X-Ray pattern of the ZA2 sample.
Page 6
The resulting nanomaterials have high crystallinity since all peaks are sharp, as reflected in Fig. 2, and no further annealing treatment was required.
The resulting nanomaterials have high crystallinity since all peaks are sharp, as seen in Fig. 2, and no further annealing treatment was needed.
Page 6
When only ethanol was used as a solvent in syntheses, the crystallite size increased at 15.8 nm for Z2 and 15.1 nm for ZA2.
When only ethanol was used as a solvent in syntheses, the crystallite size is increased to 15.8 nm for Z2 and 15.1 nm for ZA2.
Page 6
The histogram distributions of ZnO nanorods’ length are presented in Fig. 3 e)-f), and the diameter of Au nanoparticles in the ZA1 and the ZA2 nanocomposites in Fig. 3 g)-h).
The histogram distributions of ZnO nanorod lengths are presented in Fig. 3 e)-f), and the diameter of Au nanoparticles in the ZA1 and the ZA2 nanocomposites is in Fig. 3 g)-h).
Page 6
The diameters were 6- 20 nm, with an average of 11.8±0.2 nm (Fig. S1) and an aspect ratio of more than 4.
The diameters were 6- 20 nm, with an average of 11.8±0.2 nm (Fig. S1), and the aspect ratio was more than 4.
Page 8
These gold nanoparticles measured a mean value of 6.2±0.2 nm in diameter (Fig. 3h).
These gold nanoparticles are characterized by a mean value of 6.2±0.2 nm in diameter (Fig. 3h).
Page 8
The FTIR spectra, compared in Fig. 4, showed broad bands at 374 and 382 cm−1 for Z1 and ZA1 samples, respectively and at 391 and 412 cm−1 for Z2 and ZA2 samples, respectively, corresponding to the metal-oxygen (Zn-O) vibration mode.
The FTIR spectra, compared in Fig. 4, showed broad bands at 374 and 382 cm−1 for Z1 and ZA1 samples, respectively, and at 391 and 412 cm−1 for Z2 and ZA2 samples, respectively, corresponding to the metal-oxygen (Zn-O) vibration modes.
Page 8
The difference in the shape and position of the bands can be related to a change in the lattice parameters and the morphology of ZnO nanostructures [52, 53], as demonstrated by XRD and SEM images.
The difference in the shape and position of the bands can be related to a change in the lattice constants and the morphology of ZnO nanostructures [52, 53], as demonstrated by XRD and SEM images.
Page 8
The band from 1183 cm-1 was assigned to the rocking deformation of NH3+.
The band at 1183 cm-1 was assigned to the rocking deformation of NH3+.
Page 8
In the 1000-600 cm-1 domain, few bands were recorded and can be assigned to the vibrations of the backbone bonds from lysine or SDBS, peaks centred at 989 (C-C-H) 842 (C-N), 822 (C-OH), and 617 cm-1 (C-H).
In the 1000-600 cm-1 domain, few bands were recorded and they can be assigned to the vibrations of the backbone bonds from lysine or SDBS, peaks centred at 989 (C-C-H) 842 (C-N), 822 (C-OH), and 617 cm-1 (C-H).
Page 9
Suspensions of each sample in distilled water (0.1 mg/mL) were obtained as in reference [56] using an ultrasonic probe fitted with a vial tweeter. The optimised parameters 339 for sonication were found to be 10 W with amplitude adjustment at 90% for 2 minutes.
Suspensions of each sample in distilled water (0.1 mg/mL) were obtained as in [56], using an ultrasonic probe fitted with a vial tweeter. The optimised parameters 339 for sonication were found to be 10 W with amplitude adjustment at 90% for 2 min.
Page 9
Lysine-capped Au NPs had a high negative charge [58, 59] in contact with ZnO NPs, which led to changes in the electrical charge of the composite system.
Lysine-capped Au NPs had a high negative charge [58, 59] in contact with ZnO NPs, which led to a change in the electrical charge of the composite system.
Page 10
Figures 5a-b and 5c show the results of quantitative and qualitative antibacterial activity evaluation of ZnO and ZnO/Au nanomaterials using Escherichia coli (Gram-negative) and Staphylococcus aureus (Gram-positive) bacterial strains.
Figures 5a-b and 5c show the results of quantitative/qualitative evaluation of the antibacterial activity of ZnO and ZnO/Au nanomaterials using Escherichia coli (Gram-negative) and Staphylococcus aureus (Gram-positive) bacterial strains.
Page 10
The qualitative antibacterial assay employed the standard Kirby–Bauer disk diffusion method and the diameters of inhibition zones were tabulated in Table 2.
The qualitative antibacterial assay employed the standard Kirby–Bauer disk diffusion method and the diameters of inhibition zones are tabulated in Table 2.
Page 10
this result leads us to reject the null hypothesis, according to which the model is not statistically significant.
This result leads us to reject the null hypothesis, according to which the model is not statistically significant.
Page 11
The values for the ZI are higher in the E. coli bacteria test than the S. aureus test.
The values for the ZI are higher in the E. coli bacteria test than in the S. aureus test.
Page 11
Staphylococcus aureus has a cytoplasmic membrane and thick cell walls composed of several layers of peptidoglycan.
- aureus has a cytoplasmic membrane and thick cell walls composed of several layers of peptidoglycan.
Page 11
In contrast, Escherichia coli has a more complex cell wall structure, with layers of peptidoglycan between the outer membrane (lipopolysaccharide molecules with negative charges) and the cytoplasmic membrane.
In contrast, E. coli has a more complex cell wall structure, with layers of peptidoglycan between the outer membrane (lipopolysaccharide molecules with negative charges) and the cytoplasmic membrane.
Page 11
This difference in the cell wall structure and its electrical charge can facilitate the entrance of positive ions released by the ZnO into the Escherichia coli cell, which explains the better antibacterial results recorded against these bacteria strains (Fig. 5).
This difference in the cell wall structure and its electrical charge can facilitate the entrance of positive ions released by the ZnO into the E. coli cell, which explains the better antibacterial results recorded against these bacteria strains (Fig. 5).
Page 11
These electrical interactions between the two types of antibacterial material differ as ZnO are positively charged, and ZnO/Au ones are slightly negatively charged (see Table 1).
These electrical interactions between the two types of antibacterial materials differ as ZnO are positively charged, and ZnO/Au ones are slightly negatively charged (see Table 1).
Page 11
In this proposed mechanism, Staphylococcus aureus, a Gram-positive bacterium, proved to be more protected due to the thick peptidoglycan layer (up to 40 layers), which confers significant mechanical strength.
In this proposed mechanism, S. aureus, a Gram-positive bacterium, proved to be more protected due to the thick peptidoglycan layer (up to 40 layers), which confers a significant mechanical strength.
Page 11
Recently it has been demonstrated that 100 nm Au NPs can kill bacteria through mechanical deformation of the cell wall when they come in contact with the cells [63].
Recently, it was demonstrated that 100 nm Au NPs can kill bacteria through mechanical deformation of the cell wall when they come in contact with the cells [63].
Page 12
Also, when the test bacteria were put in contact with the ZnO and ZnO/Au nanomaterials, the bacterial cells produce excessive ROS, which can no longer be counterbalanced by enzymes and antioxidants, leading to bacterial damage[74].
Also, when the test bacteria were put in contact with the ZnO and ZnO/Au nanomaterials, the bacterial cells produce excessive ROS, which can no longer be counterbalanced by enzymes and antioxidants, leading to bacteria damage[74].
Page 12
The present study focused on the wet chemical-based synthesis (solvothermal and chemical reduction in solution) of zinc oxide nanostructures.
The present study was focused on the wet chemical-based synthesis (solvothermal and chemical reduction in solution) of zinc oxide nanostructures.
Page 12
L-Lysine capped Au NPs were in-situ synthesised and supported on ZnO nanorods.
L-Lysine capped Au NPs were in-situ synthesized and supported on the ZnO nanorods.
Page 12
Therefore, the ZnO/Au nanomaterials are potential as antimicrobial agents. Since the advantages such as biocompatibility, high-temperature resistance, and cost-effective synthesis further consolidate the ZnO and ZnO/Au NPs’ profile and potential, future studies will conclude the best formulations for successful implementation in the healthcare industry for the maximum benefit of the patients, professionals, and society.
Therefore, the ZnO/Au nanomaterials are potential antimicrobial agents. Since the advantages, such as biocompatibility, high-temperature resistance and cost-effective synthesis, further consolidate the ZnO and ZnO/Au NPs profile and potential, future studies will conclude the best formulations for successful implementation in the healthcare industry for the maximum benefit of the patients, professionals and society.
Response: We made all the changes that you suggested in the text and they are written in red.
Zinc nitrate hexahydrate (Zn(NO3)2∙6H2O), sodium dodecylbenzenesulfonic acid salt – SDBS (CH3(CH2)11C6H4SO3Na), potassium hydroxide (KOH), ethanol - C2H5OH (98%) L- lysine H2N(CH2)4CH(NH2)CO2H, Au (III) chloride trihydrate (HAuCl4∙3H2O), sodium borohydride (NaBH4), and Luria-Bertani broth (LB) were purchased from Sigma-Aldrich and Tryptone soy broth (TSB) from Oxoid.
Besides supplier, purity rate should be reported for each starting reagent.
Response: we introduced in the text the purity of all starting reagents.
Page 5
Diffraction planes of reflections for a single-phase wurtzite structure (hexagonal phase, space group P63mc) were identified for all the samples. The XRD patterns are in good agreement with JCPDS-card No. 36-1451.
The diffraction peak positions were well attributed to single-phase wurtzite structure (hexagonal phase, space group P63mc) for all the samples. The XRD patterns are in good agreement with JCPDS-card No. 36-1451.
Besides card number, the related original paper should be cited.
Response: we introduced in the text the reference [47].
Page 2
Different ZnO based nanostructures have been tested for antibacterial 78 response such as NPs [22, 23], nanorods [24] nanoflowers [25], nanofibres [26], Ag doped ZnO NPs [27, 28], surface decorated NPs [29], nanocomposites [30, 31], surface modified NPs [32].
For wider comparison, one other representative paper in this field could be additionally cited:
Int. Biodeter. Biodeg. 146 (2020) 104821
Response: Thank you for your suggestion. We introduced in the text the reference [24].
This manuscript is a resubmission of an earlier submission. The following is a list of the peer review reports and author responses from that submission.
Round 1
Reviewer 1 Report
The authors have evaluated the antibacterial activity of ZnO and ZnO/Au nanomaterials with different morphologies, synthesized though the solvothermal method. The authors have also demonstrated the structure, crystallinity, and morphology of ZnO and ZnO/Au nanomaterials by XRD, SEM, TEM, DLS, and FTIR spectroscopy. The ZnO and ZnO/Au nanomaterials nanomaterials have exhibited significant antibacterial effects on the Gram positive and Gram negative bacteria used. Overall, this work can inspire more material design ideas of ZnO-based nanomaterials for antibacterial application. Therefore, I would like to recommend this work to publish in Nanomaterials. Below are some comments for the authors.
1. The EDX spectrum of ZA1 of Figure 3e is too blurred to see the details. Please provide Figure 3e with higher resolution.
2. Please provide elemental percentages of EDX spectrum of Figure 3e. Furthermore, the EDX elemental mapping of ZA1 is also suggested to provide in Figure 3. The EDX elemental mapping of sample can provide its elemental distributions.
3. For antibacterial mechanism, the authors have described “Z1 sample (with smaller dimensions than Z2) showed better antibacterial performance in comparison with Z2, meaning that Z1 with large interfacial area (high surface area per unit of volume), enters the membrane more easily, resulting in superior antibacterial efficiency”. To demonstrate this hypothesis, the authors should provide experimental data to prove their point. For example, TEM image is accepted.
4. Again, the authors have claimed antimicrobial mechanism of ZnO and Au NPs could be through reactive oxygen species (ROS). The authors should provide ROS measurements of ZnO and ZnO/Au incubated with bacteria. For a research paper, the authors should provide experimental data to prove their hypothesis.
5. For the introduction “The bactericidal activity of NPs depends on composition, size, shape, crystallinity, defects, surface modification, and concentration in the culture media and the type of bacteria (Gram-positive or Gram-negative species)”, more references could be cited to broaden the introduction.
https://doi.org/10.3390/ijms20122924
Reviewer 2 Report
In the manuscript, ZnO and ZnO/Au nanomaterials have been synthesized by solvothermal method and investigated by XRD, SEM, TEM, DLS, and FTIR spectroscopy. The antibacterial effect of unmodified ZnO and ZnO/Au nanomaterials against Escherichia coli and Staphylococcus aureus was investigated. The experiments are well designed and the results are reliable. However, major revision are required before acception.
1. The authors should pay attention to the writing and the language need to be further polished. For example, “to counterbalance the decline in the development new antibiotics” in line 45-46 should be revised as “to counterbalance the decline in the development of new antibiotics”. Please revise the writing of subscripts in line 105 and other places.
2. Ag nanoparticles or nanocomposites are widely used as antibacterial agents. More references are suggested to be cited for comparison. For example, Nanocomposite Egg Shell Powder with in situ Generated Silver Nanoparticles Using Inherent Collagen as Reducing Agent; Preparation and Properties of Cellulose Nanocomposite Fabrics with in situ Generated Silver Nanoparticles by Bioreduction Method.
3. “step 1A” in line 136 should be revised as “step 1a”.
4. “Cold 0.1 mM sodium borohydride aqueous solution” in line 137 should be revised. The exact temperature of 0.1 mM sodium borohydride aqueous solution should be given for other researchers to repeat the experiments.
5. HRTEM images are suggested to be added in Fig.3 if available.
6. The authors should do some comparison with other ZnO based nanomaterials on the antibacterial activity.
7. Most of references are too old. More references published recently should be cited.
Round 2
Reviewer 1 Report
The authors have addressed all issues raised by the reviewers. Therefore, I would like to recommend this manuscript to publish as its current form in Nanomaterials.
